# Enhanced Delivery of Rose Bengal by Amino Acids Starvation and Exosomes Inhibition in Human Astrocytoma Cells to Potentiate Anticancer Photodynamic Therapy Effects

**DOI:** 10.3390/cells11162502

**Published:** 2022-08-11

**Authors:** Bianca Slivinschi, Federico Manai, Carolina Martinelli, Francesca Carriero, Camilla D’Amato, Martina Massarotti, Giorgia Bresciani, Claudio Casali, Gloria Milanesi, Laura Artal, Lisa Zanoletti, Federica Milella, Davide Arfini, Alberto Azzalin, Sara Demartis, Elisabetta Gavini, Sergio Comincini

**Affiliations:** 1Department of Biology and Biotechnology, University of Pavia, 27100 Pavia, Italy; 2Department of Medicine, Surgery and Pharmacy, University of Sassari, 07100 Sassari, Italy

**Keywords:** rose bengal, glioblastoma, photosensitizers, nanomedicine, drug delivery

## Abstract

Photodynamic therapy (PDT) is a promising anticancer strategy based on the light energy stimulation of photosensitizers (PS) molecules within a malignant cell. Among a multitude of recently challenged PS, Rose bengal (RB) has been already reported as an inducer of cytotoxicity in different tumor cells. However, RB displays a low penetration capability across cell membranes. We have therefore developed a short-term amino acids starvation protocol that significantly increases RB uptake in human astrocytoma cells compared to normal rat astrocytes. Following induced starvation uptake, RB is released outside cells by the exocytosis of extracellular vesicles (EVs). Thus, we have introduced a specific pharmacological treatment, based on the GW4869 exosomes inhibitor, to interfere with RB extracellular release. These combined treatments allow significantly reduced nanomolar amounts of administered RB and a decrease in the time interval required for PDT stimulation. The overall conditions affected astrocytoma viability through the activation of apoptotic pathways. In conclusion, we have developed for the first time a combined scheme to simultaneously increase the RB uptake in human astrocytoma cells, reduce the extracellular release of the drug by EVs, and improve the effectiveness of PDT-based treatments. Importantly, this strategy might be a valuable approach to efficiently deliver other PS or chemotherapeutic drugs in tumor cells.

## 1. Introduction

Malignant brain tumors of glial nature such as Glioblastoma (GBM, WHO grade IV) are the most aggressive type of Central Nervous System (CNS) tumors in humans [1]. Following current treatment schemes, as surgery, chemo- and/or radio-therapy [2], the disease persists and relapses in a few months after diagnosis [3]. Despite many therapeutic strategies, including first-line chemotherapeutic drugs such as temozolomide or carmustine, a significant efficacy in clinics has not yet been achieved due to disadvantageous drug pharmacokinetics and to the intrinsic and acquired resistance of the tumors to the treatments [4,5,6]. Consequently, the identification of novel strategies that might overcome the tumor cell resistance to anticancer treatments will be crucial to improve the quality of life and survival time of brain tumor patients [7].

Recently, basic research studies followed by clinical trials based on photodynamic therapy (PDT) in cancers, including GBM, have been reported [8,9,10]. In principle, PDT is effective in inducing reactive oxygen species (ROS) production within cells using a combination of light energy and photosensitizers (PS) moieties targeting oxygen molecules, resulting in the production of high levels of oxidative stress and thus inducing tumor cells toxicity. In particular, the toxic effect is mainly attributed to ROS accumulation and interaction with essential macromolecules such as proteins, unsaturated fatty, acids and cholesterols, inducing the irreversible damage of the integrity and functionality of the intracellular organelles (i.e., mitochondria, lysosomes and the endoplasmic reticulum), ultimately triggering cell death fates [11]. Importantly, the efficacy of the induced photo-toxic effects depends on the cellular uptake yield of PS as well as on their specific intracellular localization. For brain tumors, several clinical studies produced significant improvement in the performance of fluorescence-guided surgery using PS as hematoporphyrin, talaporfin sodium, 5-aminolevulinic acid (5-ALA), and metatetra (hydroxyphenyl) chlorin derivatives, while not so evident effectiveness for the prognosis of GBM patients was documented [12,13].

Among a multitude of available photoactive molecules, Rose bengal (RB, employed as a disodium salt form) is a water-soluble PS with two anionic charges in solution, which displays a high absorption in the visible region of the spectrum (around 552 nm) accomplished with a relatively high quantum yield of production of singlet oxygen species [14]. Due to its biochemical features, RB has a low efficiency to cross cell membranes and enter cells in the absence of specific carriers [15], and consequently, different RB hydrophobic derivatives (e.g., acetate or phosphate) have been developed [16]. It has been reported that hydrophilic PS are more advantageous than hydrophobic ones since they can be easily delivered intravenously for tumor targeting [17]. However, hydrophilic PS show a reduced uptake by tumor cells because the cellular transport systems in these cells are reduced compared to normal ones [18]. Nevertheless, RB demonstrated a broad spectrum of induced cytotoxicity against tumor and microbial cells [19]. In oncology, the main relevant applications of RB (namely PV-10, a 10% RB solution) were reported in the treatments of local and metastatic melanoma, both in the absence of external stimuli such as ultrasound and PDT stimuli [20]. In melanoma cells, RB was reported to induce death pathways as necrosis and caspase-dependent and-independent apoptosis [21,22,23,24]. Differently, to date, no evidence has been reported on the effect of RB in human astrocytic tumor cells as well as its use as a PS in PDT in vitro schemes.

In this contribution, we describe a novel combined experimental approach to significantly increase the capability of tumor cells, particularly astrocytomas, to internalize RB for PDT stimulation to finally trigger an anticancer response.

## 2. Materials and Methods

### 2.1. Cell Culture Conditions, Chemicals and PDT Stimulation

High-grade human astrocytoma (i.e., T98G and U373-MG) and C6 rat glioma established cell lines were obtained from the American Type Culture Collection (Manassas, VA, USA); Res186 and Res259 low-grade human pediatric astrocytoma established cell lines, and rat normal astrocytes were respectively provided by Dr. M. Bobola (University of Washington, Seattle, WA, USA) and Prof. S. Schinelli (University of Pavia, Pavia, Italy) and respectively described in [25,26]. Cells were routinely grown as monolayers at 37 °C in high-glucose (D-glucose = 4500 mg/L) Dulbecco’s modified Eagle’s medium (DMEM) supplemented with 10% fetal bovine serum (FBS) or exosome-free FBS for EVs isolation (Thermofisher, Waltham, MA, USA) and 100 μg/mL of penicillin–streptomycin (both from Euroclone, Milan, Italy) under atmosphere controlled at 5% O_2_. Cells were also cultured in starvation-induced schemes with low-glucose (D-glucose = 1000 mg/L) DMEM supplemented with 10% (FBS) or Hanks’ Balanced Salt solution (HBSS) (D-glucose = 1000 mg/L) media without FBS (both from Euroclone). 

Rose bengal sodium salt (RB, >95% purity, Sigma, St. Louis, MO, USA) was resuspended in sterile water, filtered by 0.22 µm pores and administered at different concentrations for different time intervals. To stain mitochondria in living cells, MitoTracker Green FM (Invitrogen, exc. = 488 nm) was employed as follows: cells were incubated with 100 nM dye at 37 °C for 45 min and then visualized by confocal microscope (see Section 2.8). Berberine chloride salt (BBR, >95% purity, Sigma) was resuspended in DMSO, filtered by 0.22 μm pores and administered at 10 μM concentration for 24 h and visualized by inverted fluorescent microscope (Nikon Eclipse TS100, 20× magnification; UV2A blocking filter (exc. = 355/50 nm; dichr. mirr. = 400 nm; barrier = 410 nm). Acridine orange dye (Sigma, 0.1 μg/mL for 5 min at 37 °C) was used to visualize red-fluorescent acidic autophagolysosomes by inverted fluorescent microscope (Nikon Eclipse TS100, 40× magnification, B2A blocking filter).

The exosomes inhibitor GW4869 (GW, >95% purity, Sigma) was resuspended in DMSO, filtered by 0.22 μm pores and administered at different concentrations (i.e., 0, 1, 2, 3, and 5 μM) for 24 h. Control-mock administration experiments (using the same DMSO concentration adopted for 5 μM solution) were performed to exclude DMSO-induced cytotoxicity. 

PDT stimulations for RB were performed using the Chemidoc MP instrument (Biorad, Hercules, CA, USA) at different time intervals and setting excitation/emission to 562/576 nm with intensity = 2.76 mW/mm^2^.

### 2.2. Colorimetric Cell Viability Assays

MTS assay was performed as we already described [27]. Briefly, cells were seeded at a density of 4 × 10^3^ cells/well in 96-well plates in a volume of 200 μL for 24 h and then treated with several concentrations of RB (0, 25, 50, 75, 100, 125, 150 and 175 μM for 24 h) exposed (PDT+) or not (PDT−) to 562/576 nm irradiation for 5 min. After 24 h p.t., 20 μL of Cell Titer One Aqueous Solution (Promega, Madison, WI, USA) was added in each well and incubated for 2 h at 37 °C. Then, absorbance was measured using a microplate reader (Sunrise, Tecan, Männedorf, Switzerland) at a wavelength of 492 nm. All experiments were performed in triplicate with independent assays.

### 2.3. RB Internalization Analysis

RB uptake in normal rat astrocytes and in T98G astrocytoma cells was visualized using a Nikon Eclipse TS100 inverted florescence microscope at 40× magnification (G2A blocking filter; exc. = 535/50 nm; dichr. mirr. = 577 nm; barrier = 580 nm), with Amnis ImageStreamX MkII flow cytometry (Luminex, Austin, TX, USA) using trypsinized cells (after 12.5 μM RB for 24 h). After trypsinization, cells were analyzed for bright field (Ch04) and for RB fluorescence (Ch03) at 60× magnification. RB internalization measurements were also performed using the Guava Muse Cell Flow Cytometer Analyzer (Luminex) with a laser stimulation of 532 nm and adopting a barrier filter of 576 ± 14 nm.

### 2.4. Isolation, Analysis and Fluorescent Staining of EVs

The extraction and purification of EVs (exosomes) from cell culture media containing exosome-free FBS (Invitrogen, Carlsbad, CA, USA) of T98G cells were conducted as described [28], adopting a Total Exosome Isolation kit from Culture Media (Invitrogen). In detail, culture media were collected, and one 0.5 volume of total exosome isolation reagent was added to the media and incubated at 4 °C for 20 h. Then, EVs were pelleted at 12,000 g for 60 min at 4 °C and finally resuspended into 500 μL of sterile ice-cold D-PBS (Thermofisher) and next filtered (0.22 μm). The qNano Gold instrument (Izon Science, Christchurch, New Zeeland) was employed to measure the size distribution and concentration of the isolated nanoparticles using the Tunable Resistive Pulse Sensing (TRPS) principle [29,30]. Briefly, 35 μL of purified particles were analyzed with a qNano Gold instrument using a NP200 Nanopore (Izon Science) and applying 49 mm stretch, 0.1 V, and 20 mBar parametric conditions. The calibration particles (CPC100, Izon Science) were assayed before the experimental samples under identical conditions. Size and concentrations (2000 events each) were finally determined using the qNano software provided by Izon Science (Izon Control Suite version 3.1). A rabbit anti-CD9 antibody (Cell Signaling, #13174), fluorescently conjugated using a DyLight488 labeling kit (Biorad), was used to stain EVs. CD9 conjugated DyLight488 antibody (1 μg) was centrifuged for 10 min at 17,000 g before use to eliminate aggregates and then incubated for 1 h at room temperature with EVs (~2 × 10^7^ particles) with 1% BSA (*v*/*v*). Then, CD9 stained particles were purified from unstained dyes using Exosome spin columns (MW 3000, Thermofisher) as recommended.

### 2.5. Imagestream Flow Cytometry Analysis

An Amnis ImageStream MkII instrument (Luminex) equipped with 3 lasers (100 mW 488 nm, 150 mW 642 nm, 70 mW 785 nm) (SSC) was used to assay RB cellular internalization in normal rat astrocytes and in T98G astrocytoma cells and to analyze exosomes. For RB internalization studies, cells were incubated with RB (25 or 175 μM for 24 h), analyzed by flow cytometry at 60× magnification (NA = 0.9; DOF = 2.5 μm, core size = 7 μm), and excited at 488 nm (Ch03, 480–560 nm, Ch width, 528/65 bandpass, at 10 mW). Channel 6 (745–800 nm filter) was used for scatterplot (SSC) detection, and a standard sheath fluid (D-PBS, Themofisher) was adopted in all measurements. Exosomes were analyzed with 60× magnification (core size = 7 µm) and setting the “High Gain” option. CD9+ exosomes were collected in channel 2 (480–560 nm filter), while RB+ exosomes were collected in channel 3 using channel 5 (595–650 nm filter). Channel 6 (745–800 nm filter) was used for scatterplot (SSC) detection. Standard sheath fluid (D-PBS, Themofisher) without further filtration was used in all measurements. Negative controls for EVs included detergent lysis controls; buffer controls without particles and unstained antibody samples were adopted. For nuclear staining, before trypsinization, cells were incubated at 37 °C for 15 min with DRAQ5 nuclear dye (Invitrogen, 1 μL of a 1/10 dilution into 30 mm plate) and analyzed by flow cytometry at 60× magnification, excited at 642 nm (Ch05, 642–745 nm, Ch width, 762/35 bandpass). Data analysis was performed using Amnis IDEAS software (Luminex, version 6.2). The gating strategies used are described in the Results section and Figures legends.

### 2.6. Viability, RB Internalization, Apoptosis, Autophagy, Oxidative Stress Cytofluorimetric Analysis

The viability of normal rat astrocytes and astrocytoma cells was also determined using the Guava Muse Count & Viability assay (Luminex), as described [31]. Briefly, after trypsinization and collection, cells were washed with PBS and incubated for 5 min at room temperature in the dark with 9 volumes of Count & Viability reagent. The analysis was then performed using the Guava Muse Cell Flow Cytometer Analyzer (Luminex). The same instrument was used in conjunction with the Guava Muse Multicaspase kit (Luminex) to assay caspases activation, with the Autophagy LC3-Antibody Based kit (Luminex) to quantify LC3 expression and with the Oxidative Stress kit (Luminex) for quantitative analysis of oxidative stress (ROS^+^ production), as we already described [27].

### 2.7. Immunoblotting Analysis of Proteins Expression

Whole protein extraction and immunoblotting analysis were performed as previously described [32]. In detail, T98G cell pellets were resuspended in ice-cold RIPA buffer (150 mM NaCl, 50 mM Tris-HCl pH 8.0, 1 mM Triton X-100, all from Sigma) supplemented with a Complete Mini protease inhibitor cocktail (Roche, Basel, Switzerland). Protein samples were quantified by Qubit fluorimeter, using a Protein Assay kit (Invitrogen) following the manufacturer’s instructions. Before loading in SDS-PAGE, protein extracts were boiled in Laemmli sample buffer (2% SDS, 6% glycerol, 150 mM B-mercaptoethanol, 0.02% bromophenol blue, and 62.5 mM Tris-HCl pH 6.8). After electrophoresis, proteins were transferred onto a nitro-cellulose membrane Hybond-C Extra (GE Healthcare, Milan, Italy). Membranes were blocked with 5% nonfat milk in PBS containing 0.1% Tween 20 (*v*/*v*) and incubated overnight at 4 °C with primary antibodies. The primary employed antibodies were caspase-3 (#D3R6Y) and PARP-1 (#46D11) (Cell Signaling, Danvers, MA, USA; diluted 1:2000); α-tubulin (Cell Signaling, #4967 diluted 1:6000) was used as the internal loading control. Species-specific peroxidase-labeled ECL secondary antibodies (Cell Signaling, diluted 1:4000) were employed. Protein signals were revealed by the Weststar Supernova Kit (Cyanagen, Bologna, Italy) and visualized using Chemidoc MP system (Biorad).

### 2.8. Confocal and Ultrastructural Microscopy Analysis

For confocal microscopy, T98G cells (10^4^) were grown on 25 mm diameter round coverlips in standard growing (DMEM high glucose) and then incubated for 24 h with RB (12.5 μM) and/or MitoTracker Green FM (Invitrogen, 100 nM for 45 min) in the absence of PDT stimulation. RB and mitochondria fluorescence were imaged using a LEICA TCS SP8 confocal microscope (Leica Microsystems, Wetzlar, Germany) using a 63× oil immersion objective (Leica HP PL APO CS2 63×/1.4). For the excitation, a white light laser tuned at 488 nm (for MitoTracker Green) and 545 nm (for RB) was used, and the fluorescence emissions were kept in 500–540 nm and 557–700 nm ranges, respectively. Images were acquired as a single z plane with an optical zoom of 2.5× and a format of 1024 × 1024 pixels.

For transmission electron microscopy (TEM) analysis, T98G cells (2 × 10^6^) were grown in DMEM medium in 90 mm plates. Cells in standard growing (DMEM high glucose), in HBSS starvation alone (30 min), after RB (0.75 μM) and GW4869 (5 μM) HBSS starvation, and in the absence or presence of PDT stimulation (2 min at 562/576 nm) were harvested by centrifugation at 800 rpm for 5 min and fixed with 2.5% (*v*/*v*) glutaraldehyde in PBS for 2 h at room temperature as described [27]. Cells were then rinsed in PBS (pH 7.2) overnight and post-fixed in 1% aqueous OsO_4_ for 1 h at room temperature. Cells were pre-embedded in 2% agarose in water, dehydrated in acetone, and finally embedded in epoxy resin EM-bed812 (Electron Microscopy Sciences, Hatfield, PA, USA). Ultrathin sections (60–80 nm) were collected on nickel grids and stained with uranyl acetate and lead citrate. The specimens were observed with a JEM 1200 EX II (JEOL, Peabody, MA, USA) electron microscope, equipped with the MegaView G2 CCD camera (Olympus OSIS, Tokyo, Japan) and operating at 120 kV. The morphology of mitochondria (at least 10 for each sample) was then analyzed by two independent evaluators.

### 2.9. Statistical Analysis

The data were analyzed using the statistics functions of the MedCalc statistical software version 18.11.6. (http://www.medcalc.org (accessed on 15 February 2021)). The ANOVA test differences were considered statistically significant when *p* < 0.05.

## 3. Results

A concentration-range kinetics of RB administration (0 to 175 μM) was initially performed in different cell lines, including normal rat astrocytes as well as human low- (i.e., Res186) and high-grade astrocytoma (U373-MG and T98G) cells. Cells were also exposed (PDT+) or not (PDT−) for 5 min at 562/576 nm light irradiation. After 24 h of incubation, MTS viability assays were performed. As reported in Figure 1, only the highest RB doses (i.e., 150 and 175 μM) accomplished with PDT stimulation induced a significant decrease in the tumor cells’ viability. Noticeably, RB and/or PDT did not significantly affect normal astrocytes’ viability compared to the highest toxicity scored in T98G cells (Appendix A).

To evaluate if the viability differences scored between normal cells (i.e., astrocytes) and tumor ones (i.e., T98G) were due to the different RB uptake efficiencies that sustained PDT effects, fluorescence microscope examinations and flow cytometry internalization measurements were performed on the lowest (RB = 25 μM) and highest (RB = 175 μM) doses (Figure 2A,B). Both assays indicated that T98G cells internalized RB with higher efficiency compared to normal astrocytes, showing cytofluorimetric internalization indexes of 2.96 and 1.44, respectively, following RB administration at 25 μM. 

Following RB (25 μM) internalization after 24 h p.t., as reported in Figure 3, RB was mostly localized within mitochondria, according to RB and MitoTracker (Invitrogen) staining and revealed by confocal microscope examinations in living T98G cells. 

To improve contemporary RB uptake with the aim to reduce its dosage, different metabolic restrictions were assayed as the use of low-glucose growing conditions (i.e., D-glucose = 1000 mg/L) or the HBSS without amino acids and compared against a standard high-glucose (i.e., D-glucose = 4500 mg/L) growing condition. RB (12.5 μM) was administered and microscopically visualized after 2 h of incubation; in addition, the corresponding cells were trypsinized and analyzed by cytometry to evaluate RB intensity. As reported in Figure 4A, even through optical microscope examinations, amino acids starved T98G cells displayed higher RB internalization indexes compared to low-glucose and standard growing conditions. Nearly identical results were scored for normal astrocytes and astrocytoma cells (i.e., Res186 and U373-MG) (data not shown). The high efficiency of RB internalization following amino acids starvation was confirmed by the cytofluorimetric evaluation of RB intensity by Guava Muse Cell Analyzer (Luminex). Among the investigated cell lines (i.e., rat normal astrocytes and C6 rat glioma; human low grade astrocytomas Res189 and Res259; human high grade astrocytomas U373-MG and T98G), the most significant normalized difference between RB administration in standard condition and in amino acids starvation was reported for T98G (RB = 1.0 and RB = 25.7, respectively) (Figure 4B).

To further confirm that the amino acids starvation protocol allows us to improve the uptake of compounds within the cell and to verify that this principle is not exclusive to RB, T98G cells were incubated for 30 min in standard medium or in amino acids depleted HBSS with a different PS, such as berberine (BBR) at a concentration of 10 μM. As shown in Appendix A, soon after the scheduled interval, starvation induced a marked uptake of BBR compared to the control. In addition, after PBS washing and standard medium replacement, after 24 h p.t., a reduction of proliferation and cytotoxicity was scored, confirming the greater extent of BBR internalization following starvation conditions. 

Following starvation, RB (12.5 μM) uptake was monitored at different time intervals by fluorescent microscope in normal rat astrocytes and in T98G cells. RB, as documented in Figure 5A, quickly localized at plasma membranes at 3 min p.t. and next internalized within cells in 10 min, with higher efficiency in T98G cells as previously documented. However, after 2 h of amino acids starvation in HBSS media and by restoring standard growing conditions (i.e., high glucose media), following 24 h of further incubation, fluorescence and optical microscope analysis indicated an evident reduction of intracellular RB signals. These evidences were confirmed by cytofluorimetric measurements of RB intensity. While astrocytes did not highlight significant differences in the normalized RB intensity comparing 10 min and 24 h intervals, in T98G cells, the RB-normalized fluorescence intensity shifted from 4.1 (10 min) to 1.7 (24 h) with an overall reduction of 41.9% of RB intracellular intensity.

To explain the RB intracellular decrease in T98G cells, we primary speculated on the possibility of the activation of the autophagy process inducing RB clearance in starvation to normal growing transition. Thus, the expression of the autophagosome-associated marker LC3 was evaluated in T98G cells at 2 h following HBSS starvation and at 24 h p.t. after a standard growing condition and compared to cells treated with the autophagy inducer Rapamycin (1 μM for 24 h in standard growing conditions, as described in [31]) by cytometry using a Muse Autophagy LC3-Antibody Based kit (Luminex). In parallel, the same samples, after starvation and growing reconditioning were stained with Acridine orange dye (Sigma, 0.1 μg/mL for 5 min at 37 °C) to visualize red-fluorescent acidic autophagolysosomes. As reported in Appendix A, no significant differences in LC3 intensity between amino acids starvation and reconditioning conditions were scored by cytofluorimetry and for the presence of red-acidic autophagy vesicles after fluorescent microscopy evaluations. Consequently, we hypothesized that the RB fluorescent reduction in T98G cells might be related to the activation of exocytosis through extracellular vesicles (EVs) release processes as a consequence of the rescue of metabolic growing conditions. To verify this hypothesis, a strategy based on the use of the commonly pharmacological agent GW4869 (Sigma), which inhibits EVs generation, blocking ceramide-mediated inward budding of multivesicular bodies (MVBs) and the release of mature EVS from MVBs, was considered. Firstly, we assayed the capability of GW4869 to reduce EVs, specifically exosomes, release in T98G cells through a dose kinetics of GW4869 administration (at 0–1–3–5 μM). Exosomes were collected in standard growing media (supplemented with exosomes-free FBS) using a Total Exosomes Isolation kit (Thermofisher) and analyzed in size and concentration by means of TRPS using a qNANO Gold instrument (Izon), as we previously reported [25,32]. As shown in Figure 6, GW4869 showed a dose-dependent effect in the reduction of exosomes concentration in the media, with in particular 7.9 × 10^8^ particles/mL in the absence of GW4869 and 0.8 × 10^8^ particles/mL in correspondence of the highest dose (i.e., 5 μM), thus producing a 9.9-fold reduction in exosomes release.

We next evaluated GW4869 doses (0–1–3–5 μM) effects on normal astrocytes and T98G viability in standard growing conditions at 24 h p.t. using the cytofluorimetric Muse Count & Viability kit (Luminex). As a control of the possible solvent toxicity, DMSO (mock) was tested at the corresponding volume used to assay the highest GW4869 dose (i.e., 5 μM) and compared to GW4869 untreated cells (NT). Cytofluorimetric analyses along with optical microscope evidence did not report significant cytotoxicity of all GW4869 doses nor for mock controls (Appendix A). 

In order to verify that RB-treated cells, after amino acids starvation and rescuing of the growing conditions, activated RB clearance by exosomes release, T98G cells (7.5 × 10^6^ in two 90 mm plates) were incubated with 12.5 μM RB under HBSS starvation, which was followed by standard growing conditions (in the presence of exosomes-free FBS) for 24 h. Exosomes were then isolated from cell culture media and split into two identical aliquots. The former was not further treated, while the latter was separately stained with 1 μg of a rabbit anti-CD9 primary antibody (Cell Signaling, #13174) fluorescently conjugated using a DyLight488 labeling kit (Biorad). After excess dyes removal by Exosome Spin Purification columns (MW 3000, Invitrogen), the exosomes aliquots were analyzed by flow cytometry (Amnis Imagestream, Luminex). The results documented that the isolated particles had RB cargoes (first aliquot, R3 gate), showed positivity for the CD9 exosome marker (second aliquot, R4 gate), and thus represented exosomes-like vesicles (Figure 7).

To further assess one of the typical feature of exosomes as their cellular spontaneous internalization, 10^7^ purified CD9-stained and RB containing particles, isolated from T98G cells incubated for 24 h with RB (12.5 μM), were directly administered to growing T98G cells (10^4^ cells into a 30 mm plate). After 24 h of incubation, CD9+ and RB+ spots were detected within the cytoplasm as revealed by fluorescent microscopy and flow cytometry analysis (Appendix A). 

After this evidence, experimental variables were analyzed to set up a protocol in order to (a) increase RB uptake by amino acids starvation, thus reducing the RB molar amount of administration; (b) pharmacologically block the extracellular release of RB by GW4869; and (c) identify suitable PDT time-interval conditions to efficiently stimulate RB photo-activation in astrocytoma cells vs. normal astrocytes. Primarily, RB (12.5 μM) internalization during HBSS starvation was monitored in living astrocytes and T98G cells at different time points (i.e., 10–20–30–60–90–120 min p.t.) by cytofluorimetric evaluations. As reported in Figure 8, the highest difference between normal rat astrocytes and T98G cells in RB internalization was scored after 30 min of HBSS starvation (T98G scored a 3.84 fold difference in RB intensity compared to astrocytes; 6.11 vs. 1.89, respectively), while for longer starvation intervals, the graphs showed similar trends.

Next, RB concentrations at sub-micromolar ranges (i.e., 0.50 and 0.75 μM) with or without GW4869 (5 μM) were assayed in a selected starvation condition (i.e., HBSS medium for 30 min) combined with different PDT exposition times (i.e., 0–2–5 min at 562/576 nm) in T98G cells to evaluate differences in viability by optical microscope examinations and cytofluorimetric analysis using the Guava Muse Count & Viability kit (Luminex). The results, collected after a standard growing reconditioning for 24 h, indicated in particular that the condition of 2 min of PDT irradiation along with RB 0.75 μM and GW4869 5 μM induced a significant decrease in viability (i.e., a reduction of 54.75%) compared to not-GW4869 counterpart (mean viability% = 37.40 and 82.95, respectively) (Figure 9). Noticeably, the combination of the two assayed RB doses with that of GW4869, in absence of PDT stimulation, did not alter cells’ viability. Similarly, both RB doses, in absence of exosomes inhibitor and PDT stimulation, did not affect viability rates. The longest PDT exposition at 5 min, even producing significant viability differences, was not further considered to avoid excess of induced toxicity by irradiation in normal cells. 

The established protocol, RB = 0.75 μM administered for 30 min in HBSS starvation, followed by PDT exposition of 2 min at 562/576 nm, administration of GW 5 μM, in the presence of reconditioning standard medium for 24 h, was comparatively analyzed for viability in normal rat astrocytes and T98G cells. As reported in Figure 10, astrocytes were weakly affected by the combined treatment scheme with an average reduction of 13.04% in cytofluorimetric viability compared to the most sensible T98G cells (59.02%), which was also confirmed by optical microscope evaluations at 24 h p.t. 

Subsequently, the molecular pathways involved in the decrease in viability were investigated in T98G cells. In particular, the induction of apoptosis was analyzed by cytometry by means of the Guava Muse Multicaspase kit (Luminex). As a result, the highest mean score of caspases activation was reported in HBSS RB + GW+ PDT sample (38.90%), compared to RB + GW one (15.15%). Importantly, single treatments (i.e., starvation by HBSS, PDT exposition, RB or GW administration) did not produce significant differences in caspases activation (Figure 11). 

To better discern the effect of the photo-activation scheme within T98G cells, the cytometric Guava Muse Oxidative Stress assay (Luminex) was employed to simultaneously determine the count and percentage of cells undergoing oxidative stress based on the intracellular detection of superoxide radicals (ROS) using dihydroethidium, which is a well-characterized reagent extensively used to detect ROS. As schematized in Figure 12, ROS-positive (ROS^+^) cells in the combined scheme with PDT (HBSS, RB, GW, PDT) evaluated at 24 h p.t. (i.e., ROS^+^ = 17.50%) was more than 10-fold compared to normal growing (DMEM; ROS^+^ = 1.00%), starvation only (HBSS; ROS^+^ = 0.77%) and with the combined treatment without PDT stimulation (HBSS, RB, GW; ROS^+^ = 1.60%).

Ultrastructural analysis by Transmission Electron Microscopy (TEM) conducted in untreated T98G cells grown in standard conditions without or with PDT stimulation (2 min at 562/576 nm) and following HBSS starvation (30 min) with RB (0.75 μM) and GW4869 (5 μM), again, without or with PDT stimulation, revealed morphological difference in mitochondria (Figure 13). In detail, HBSS, RB and GW4869 combined treatment revealed partial cristae disarrangement, cristolysis with a reduction in the mitochondrial matrix density and of the integrity of the outer membranes of the organelles; these degradative features were further increased following PDT exposition. Importantly, PDT stimulation per se did not produce significant mitochondria damages.

Basing on these evidence, the expression of key proteins involved in the apoptosis pathway such as Caspase 3 and PARP1 was investigated in T98G cells using immunoblotting. In particular, the expression of the proteins was compared following single and combined treatments in both standard and in HBSS starvation conditions, with or without PDT stimulation (2 min at 562/576 nm). As highlighted in Figure 14, in agreement with cytofluorimetric caspases data, only the combined sample, i.e., RB = 0.75 μM starved for 30 min in HBSS medium, exposed for 2 min at 562/576 nm, administered with GW4869 = 5 μM and followed by a reconditioning of 24 h in high glucose media, exhibited cleaved Caspase 3 and PARP1 fragments. Of note, single or combined treatments in high-glucose media or in HBSS alone did not produce protein cleavages. 

According to the indication of an apoptotic involvement, flow cytometric analysis of nuclear fragmentation was performed in T98G cells, subjected to standard growing conditions, in HBSS starvation alone (30 min), after RB (0.75 μM) and GW (5 μM) HBSS starvation, in the absence or presence of PDT stimulation (2 min at 562/576 nm). As reported in Appendix A, the combined PDT treatment produced the highest percentage of nuclear fragmented cells (i.e., 10.97%) compared to not-PDT combined (1.92%), starvation only (1.18%) and normal growing conditions (1.36%), underlining the inducing effect of PDT in producing DNA cleavage.

To quantify the autophagy activity associated with RB, GW4869 and PDT protocol, the measurement of the autophagy flux was performed by flow cytometry using a Guava Autophagy LC3 Antibody-Based Detection Kit in T98G cells in standard growing (DMEM), in HBSS starvation alone (30 min), after RB (0.75 μM) and GW (5 μM) HBSS starvation, in the absence or presence of PDT stimulation (2 min at 562/576 nm). Before cytofluorimetric analysis, all samples were treated for 4 h with Autophagy Reagent A to prevent the lysosomal degradation of LC3 according to the manufacturer. As reported in Appendix A, the treatments did not produce a significant increase in LC3 expression compared to the positive control, i.e., T98G cells incubated with Rapamycin (1 μM for 4 h).

## 4. Discussion

In this contribution, we have proposed a novel in vitro approach to induce an efficient internalization of RB into human astrocytoma cells to enable PDT induction of cytotoxicity.

We have primarily demonstrated that RB uptake was quite inefficient both in normal rat astrocytes and in human astrocytoma cells of different malignancy grades. This evidence is in agreement with previous results reporting that water-soluble RB as sodium salt had a low efficiency to cross cell membranes and entering within cells in the absence of a specific carrier, comparing to RB hydrophobic derivatives (e.g., acetate or phosphate) [15,16]. It is well known that the poor permeability of drugs reduces their bioavailability and thus limits clinic applications; therefore, it is mandatory to develop suitable delivery systems for achieving effective the intracellular uptake of drugs. 

In our experiments, only relatively high RB doses (>150 μM) were useful to induce PDT-mediated cytotoxic effects, particularly in high-grade astrocytoma cells. Based on this evidence, different growing conditions with RB administration in high-, low-glucose and in amino acids-depleted media were tested. Of note, even a short-term starvation condition, using HBSS media without amino acids, resulted in a pronounced internalization of RB with increasing trends, particularly in high-grade astrocytoma cells. The rationale of this behavior, showed in different cancer models, is that tumor cells evolve in a rich nutrient environment that sustains increased glycolysis activity and protein biosynthesis [33,34]. Generally, when these cells are subjected to nutrient deprivation, they activate internalization survival macropinocytosis and non-macropinocytosis-dependent mechanisms [35]. It was reported that macropinocytosis is mainly regulated through the expression of RAS oncogenes and growth factor receptors and negatively regulated by the mammalian target of rapamycin complex 1 (mTORC1) [36]. Therefore, the activation of macropinocytosis was proposed as a promising anticancer strategy to increase drug uptake [37]. It is well known that tumor cells are strongly dependent on the exogenous supply of amino acids, both essential and non-essential types [38]. When cancer cells are induced in an amino acids starvation condition, they initially activate a homeostatic response based on the up-regulation of membrane transporters [39]. These, in turn, are useful in the response to stress conditions through the increased uptake of extracellular potential nutrients from alternative sources, even before starting catabolic survival pathways based on the digestion of intracellular components [40]. It has been widely documented that amino acids interfering strategies can be performed by inhibition of their transporters, by blocking the biosynthesis of amino acids or by external depletion of amino acids in starvation conditions [41]. In particular, a time-prolonged amino acids starvation induced in cancer cells the inhibition of protein synthesis, suppressing growth, or activate programmed cell death pathways [42], specifically caspase-dependent apoptosis or autophagic processes [43,44]. Differently, a suboptimal amino acids depletion resulted in a cellular quiescence state rather than a pronounced activation of cell death pathways [39]. Thus, accordingly, we induced a short-term amino acids starvation condition that increased significantly RB extracellular uptake. In particular, the kinetics of RB internalization assays, performed following 30 min of starvation, produced a significant yield difference between normal rat astrocytes and human highly malignant astrocytoma cells (specifically T98G), without affecting the viability and activation of cell death processes. However, after the short-term amino acids starvation interval, by rescuing of the standard growing conditions, normal and, in particular, astrocytoma cells exhibited a marked decrease in RB intracellular content. We have therefore demonstrated, using flow cytometric analysis, that astrocytoma cells adopted a pro-survival mechanism based on the extracellular release of RB through EVs as exosomes rather than a degradation of RB by autophagy process. 

EVs, particularly exosomes, are lipid bilayer membrane nanovesicles of endosomal origin, exhibiting different contents of proteins, lipids and nucleic acids depending on the cell of origin as well as their growing conditions [45]. Exosomes are crucial in intercellular communication, playing critical roles in both physiological and pathological contexts [46]. In cancer contexts, exosomes can influence the tumor spread through the intercellular transfer of oncogenic molecules [47,48] and/or conferring the resistance of malignant cells toward anticancer drugs through the packaging and the next extracellular release of the therapeutic cargoes [49]. Therefore, different strategies based on pharmacological blockage or inhibition of the exosomes machinery, in synthesis and release, have been proposed as cancer therapeutic advanced strategies [50,51,52]. To date, several pharmacological inhibitors of exosomes release, as neticonazole, ketotifen, cannabidiol and GW4869, have been successfully tested in different cells [53]. Among these compounds, GW4869 is a cell-permeable, symmetrical dihydroimidazolo-amide acting as a potent, specific, non-competitive inhibitor of membrane neutral sphingomyelinase, resulting in exosomes release inhibition [54]. Luberto and collaborators previously reported [55] a protective effect by the small molecule neutral sphingomyelinase inhibitor GW4869 in MCF7 breast cancer cells reducing nuclear condensation, caspase activation, PARP degradation and trypan blue uptake after apoptotic induction by tumor necrosis factor. However, according to our results, GW4869 (employed at 2–4 fold reduced micromolar concentration) seemed to not exert this protective behavior to counteract apoptotic induction produced by the combination of RB uptake and PDT stimulation in different tumor cells as astrocytoma ones.

According to the “Minimal information for studies of extracellular vesicles 2018 (MISEV2018) guidelines” [56], it is important to identify compounds concentrations, i.e., GW4869, that induce non-toxic off-target effects to the cells to be investigated. Based on these indications, while demonstrating the effectiveness of GW4868 in reducing exosomes release, we have documented its non-toxic effect in both normal astrocytes and in human astrocytoma cells (particularly T98G ones). We have therefore set up a short-term amino acids starvation protocol to efficiently vehicle RB into astrocytoma cells, resulting in a sub-micromolar (i.e., 0.75 μM) RB administration, which is able to induce viability effects. Of note, the increase in drug accumulation in cancer cells is an important strategy to increase their vulnerability to treatment options [57,58]. Basing on RB spectrum (i.e., excitation = 559 nm; emission = 571 nm), we have exposed T98G cells as the most effective in RB starvation-induced uptake to RB administration and exosomes inhibition by GW4868 (5 μM), to different time intervals of photo-stimulation at 562/576 nm wavelength. Cytofluorimetric results indicated that the minimum time interval to produce a significant reduction of viability in T98G cells compared to normal rat astrocytes in a PDT scheme was 2 min exposition. Overall, the established protocol induced a significant decrease in the viability of T98G cells after PDT exposition while not affecting normal rat astrocytes.

It is well established that the cytotoxic effect of a PS depends on the intracellular localization [59]; furthermore, PS able to target mitochondria organelles are efficient cytotoxic inducers [60]. Due to the reported RB accumulation in mitochondria, we have primarily supposed that the RB administration protocol might affect the apoptotic process. Apoptosis, largely involved in PDT schemes, is triggered by different pathways, initiated by the activation of a family of specific cysteine aspartyl-specific proteases collectively known as caspases [61,62]. Accordingly, we have reported by immunoblotting and cytometric analysis that only PDT stimulation, following RB starvation and exosomes inhibition, produced a prominent activation of apoptosis through the generation of cleaved Caspase 3 and PARP1 fragments in T98G cells, exhibiting an increase in Caspases+ cell population, production of ROS^+^ species and fragmentation of nuclear DNA. The activation of apoptosis following RB administration, even with different dosage schemes, was similarly reported in neuroblastoma cells [63], in melanoma following UV-A and B exposures [64], in ovarian and in breast cancer cells [22,65], while, to date, no studies have been documented on RB effects in human astrocytoma cells. Previously, it was only reported, in C6 rat glioma cells, a marked apoptotic effect of RB acetate, following a 530 nm PDT stimulation [66]. On the other hand, our results indicated that for single or combined treatments, the autophagy process was not significantly involved as revealed by cytofluorimetric analysis of the flux based on LC3 expression through the blockage of lysosomal degradation [67].

To date, the reported study presents translational limitations regarding the lack of a direct application of the experimental protocol in in vivo models of astrocytomas and, in particular, to assess RB pharmacokinetics and tissues bio-distribution. To this purpose, in order to fill these important aspects, in vivo studies are scheduled to determine the possibility that the experimental scheme with PDT stimulation by optical nanofibers devices could represent an effective therapeutic strategy for the treatment of malignant gliomas.

In conclusion, this work investigated, for the first time, the cellular and molecular effect of RB and PDT schemes in human astrocytoma cells. Another novelty of this contribution is the development of in vitro experimental conditions that reduced significantly the RB dosage by inducing a short-term amino acids starvation and a concomitant blockage of the release of exosomes with RB cargo by the tumor cells, resulting in a reduction of the PDT stimulation interval necessary to induce cytotoxicity. The next experimental steps will be directed to evaluate the possibility to apply this protocol to other PS or in general to other chemotherapeutic drugs, to increase their cellular uptake and efficacy in different tumor contexts.

## Figures and Tables

**Figure 1 cells-11-02502-f001:**
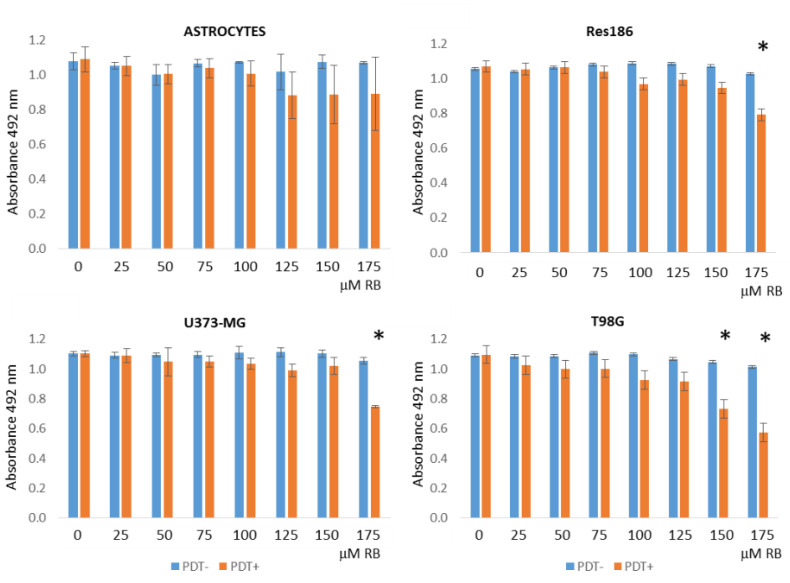
Cell viability MTS assays following RB and PDT treatments. Low- (i.e., Res186) and high-grade (U373-MG and T98G) human astrocytoma cells lines and normal rat primary astrocytes were treated with different concentrations of RB (0, 25, 50, 75, 100, 125, 150 and 175 μM for 24 h) exposed (PDT+) or not (PDT−) to 562/576 nm irradiation for 5 min. Colorimetric MTS analysis, measured as absorbance at 492 nm, was carried out after 24 h p.t. All experiments were performed in triplicates with independent assays. Asterisks indicated statistical significance compared to related PDT- cells (*p* < 0.05, Anova one-way; SD are highlighted as vertical bars).

**Figure 2 cells-11-02502-f002:**
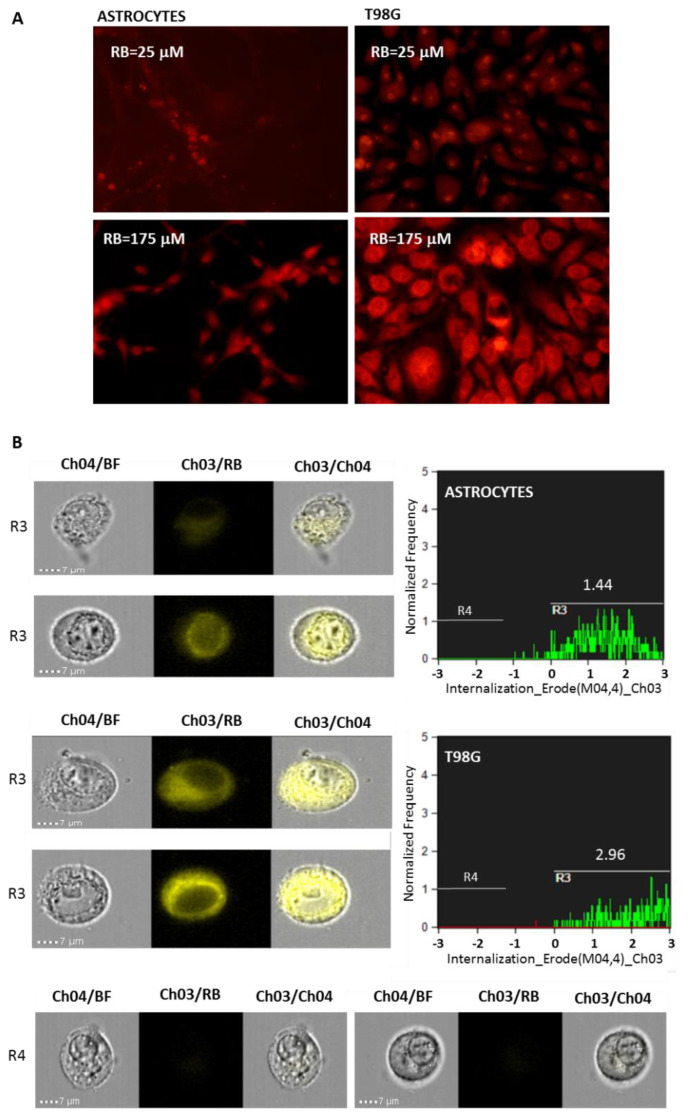
RB internalization analysis in normal rat astrocytes and in T98G astrocytoma cells. (**A**) Cells were incubated with RB (25 or 175 μM for 24 h) and visualized using a Nikon Eclipse TS100 inverted florescence microscope at 20× magnification (G2A blocking filter; exc. = 535/50 nm; dichr. mirr. = 577 nm; barrier = 580 nm). (**B**) Amnis ImageStream flow cytometry analysis of RB (25 μM) internalization in normal rat astrocytes and in T98G cells analyzed after 24 h p.t. Trypsinized cells were analyzed for bright field (Ch04) and for RB fluorescence (Ch03) at 60× magnification. Percentages of focused cells (*n* = 10,000) with RB internalization spots and Internalization Erode median parameters are reported. RB internalization was analyzed by the “Internalization” wizard with the following gating strategy: single cells were gated (using Area/Aspect Ratio Intensity, R1, not shown); R1 was then gated for focused cells using the “Gradient-RMS” feature (R2 gate, not shown), and finally, Ch3 intensity was measured with “Internalization” feature and an “Erode” mask applied (gates R3 and R4). Examples of cell images of R3 and R4 gates are shown.

**Figure 3 cells-11-02502-f003:**
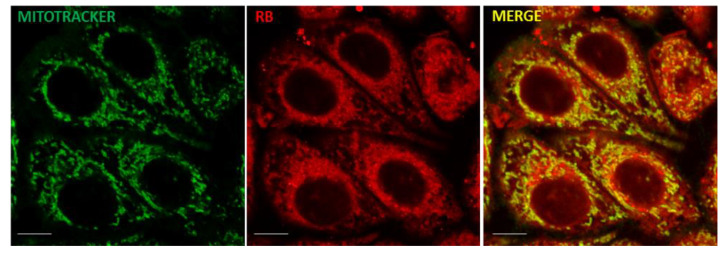
RB and mitochondria localization. T98G cells were incubated for 24 h with RB (25 μM) and analyzed after 24 h p.t. Before confocal microscopy examination, cells were treated with 100 nM MitoTracker Green FM (Invitrogen) for 45 min at 37 °C and then visualized using a Leica TCS SP8 confocal microscope at 63× magnifications (RB exc. = 545 nm; emiss. = 557/700 nm; MitoTracker: exc. = 488 nm; emiss. = 500/540 nm). Scale bars = 5 μm.

**Figure 4 cells-11-02502-f004:**
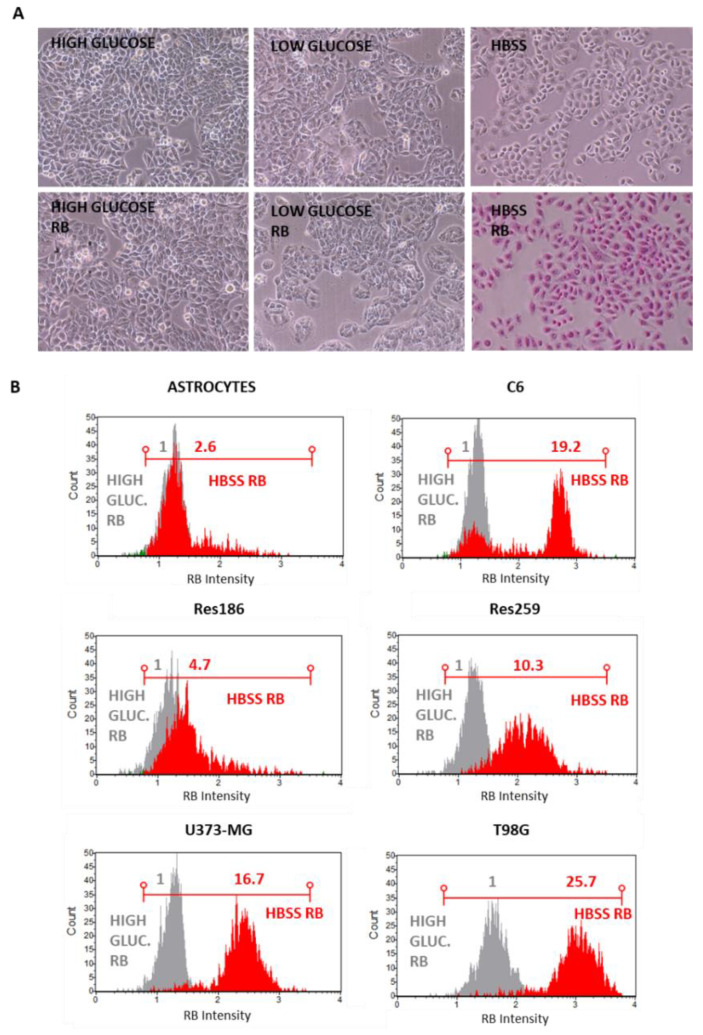
Amino acids starvation-induced uptake of RB in different cell lines. (**A**) T98G cells were incubated or not for 24 h with RB (12.5 μM) in standard DMEM high glucose (D-glucose = 4500 mg/L), DMEM low-glucose (D-glucose = 1000 mg/L) or HBSS amino acids-depleted media and visualized after 24 h p.t. using a Nikon Eclipse TS100 inverted microscope at 10× magnification. NT = RB untreated cells in standard high glucose medium. (**B**) Similar RB incubation experiments were performed in normal rat astrocytes, rat C6 glioma, low- (Res186-Res259) and high-grade astrocytoma cells (U373-MG and T98G). After incubation, cells were trypsinized and analyzed for RB fluorescence intensity using the Guava Muse Cell Analyzer (Luminex, Laser = 532 nm; Filter = 576 ± 14 nm). Each plot was obtained analyzing 5000 cells and normalized to the corresponding RB intensity in standard high glucose (RB = 1) and compared to fold differences in HBSS RB conditions.

**Figure 5 cells-11-02502-f005:**
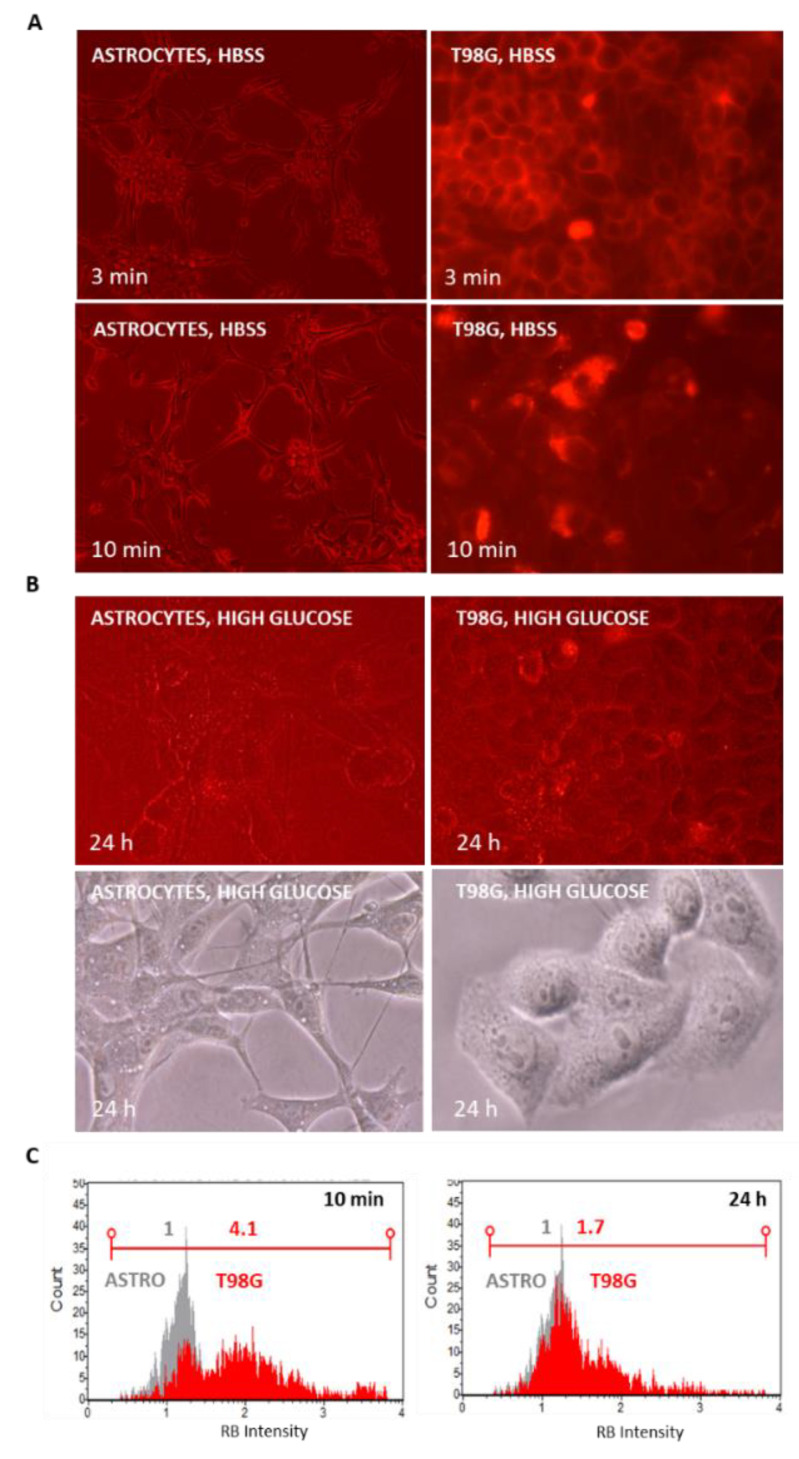
RB intracellular measurements in normal rat astrocytes and T98G cells following amino acids starvation to standard growing conditions. (**A**) Cells were incubated with RB (12.5 μM) in HBSS media and visualized at 3 and 10 min p.t. using a Nikon Eclipse TS100 inverted fluorescence (20×) or optical microscope at 40× magnification (G2A blocking filter; exc. = 535/50 nm; dichr. mirr. = 577 nm; barrier = 580 nm). (**B**) After reconditioning for 24 h with standard high glucose media, cells were visualized as above under fluorescence and optical microscope. (**C**) In parallel, trypsinized cells (*n* = 5000 each) were analyzed in RB intensity using a Guava Muse Cell Analyzer (Luminex, Laser = 532 nm; Filter = 576 ± 14 nm), comparing intensity plots at 10 min vs. 24 h (astrocytes in gray; T98G in red). RB intensity values are normalized to astrocytes RB at 10 min (RB = 1).

**Figure 6 cells-11-02502-f006:**
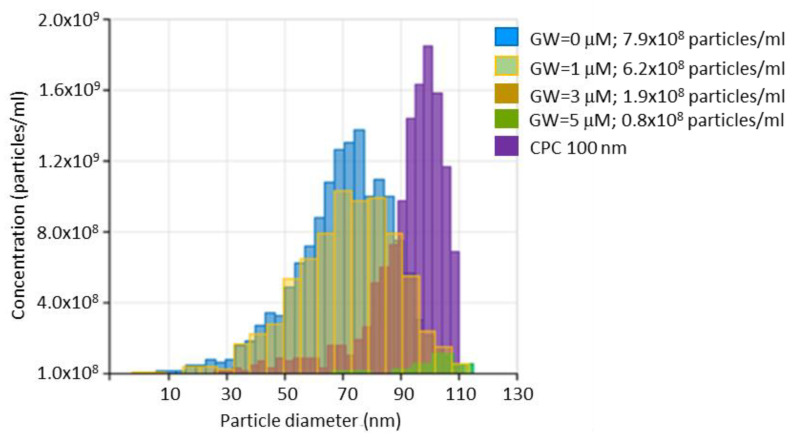
Exosomes size and concentration analysis following GW4869 treatments. T98G cells (10^6^ cells into 60 mm plates) were incubated for 24 h with different GW4869 (GW) concentrations (0–1–3–5 μM) in high-glucose media (supplemented with exosomes-free FBS). Exosomes were isolated with a Total Exosomes Isolation kit (Thermofisher) and analyzed in size and concentration by means of a qNANO Gold instrument (Izon, NP80 Nanopore; stretch = 49 nm; voltage = 0.1 V; pressure = 20 mbar). Calibration particles (CPC100, Izon, in purple) were used as size control. Particles concentrations/mL and diameters (nm) are reported.

**Figure 7 cells-11-02502-f007:**
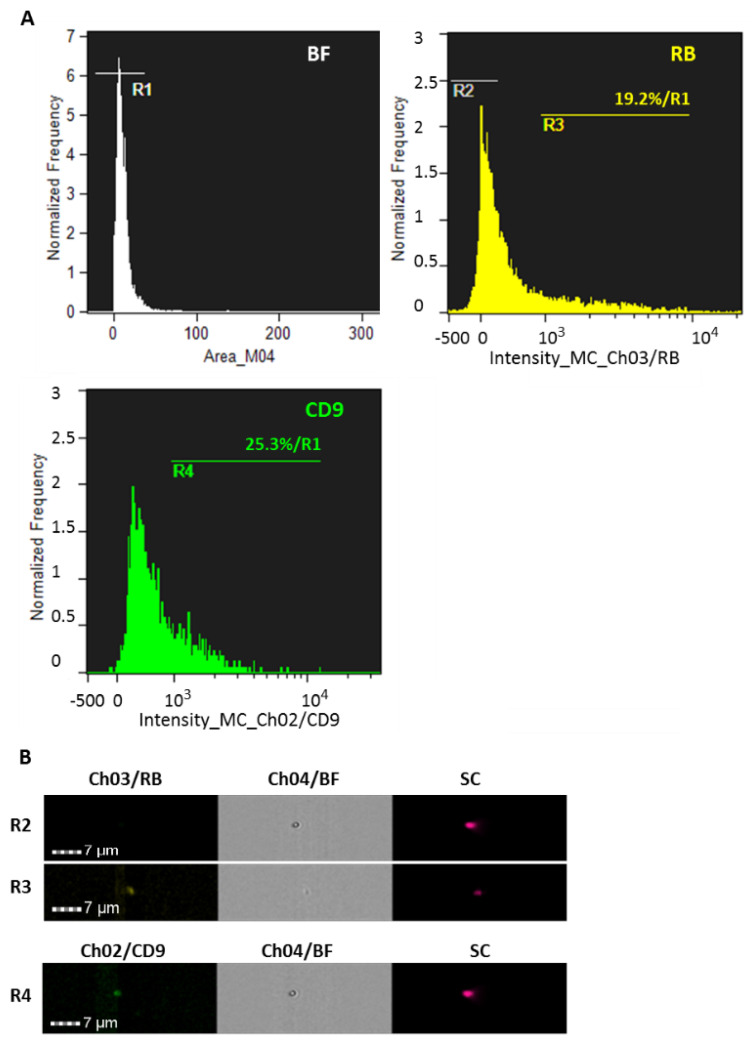
Flow cytometric characterization of exosomes. (**A**) T98G cells were incubated with 12.5 μM RB under HBSS starvation, which was followed by standard growing conditions (in the presence of exosomes-free FBS) for 24 h. Exosomes were then isolated from cell culture medium and split into two aliquots of 100 μL. The first aliquot was not further treated, while the remaining one was separately stained with 1 μg of a rabbit anti-CD9 primary antibody (Cell Signaling, #13174) fluorescently conjugated using a DyLight488 labeling kit (Biorad). After excess dyes removal by Exosome spin columns (MW 3000, Thermofisher), the exosomes aliquots were analyzed by flow cytometry (Amnis Imagestream, Luminex). High gain mode acquisition with 60× magnification was applied to plot particles distribution according to their size (Area_M04), thus defining the R1 gate. Plot distribution of RB intensity (Ch03) was reported, where R2 and R3 gates represented negative to faint fluorescence signals, and detectable RB particles, respectively. Plot distribution of CD9 intensity (Ch02) was reported, with the R4 gate representing florescent detectable CD9+ particles. For each plot, percentages of florescent detected particles on the R1 gate were reported. (**B**) Examples of image galleries within the indicated gates–channels are reported. Additional images are shown in Appendix A. BF = bright field; SC = size scatter. A total of 25,000 particles for each analysis was acquired.

**Figure 8 cells-11-02502-f008:**
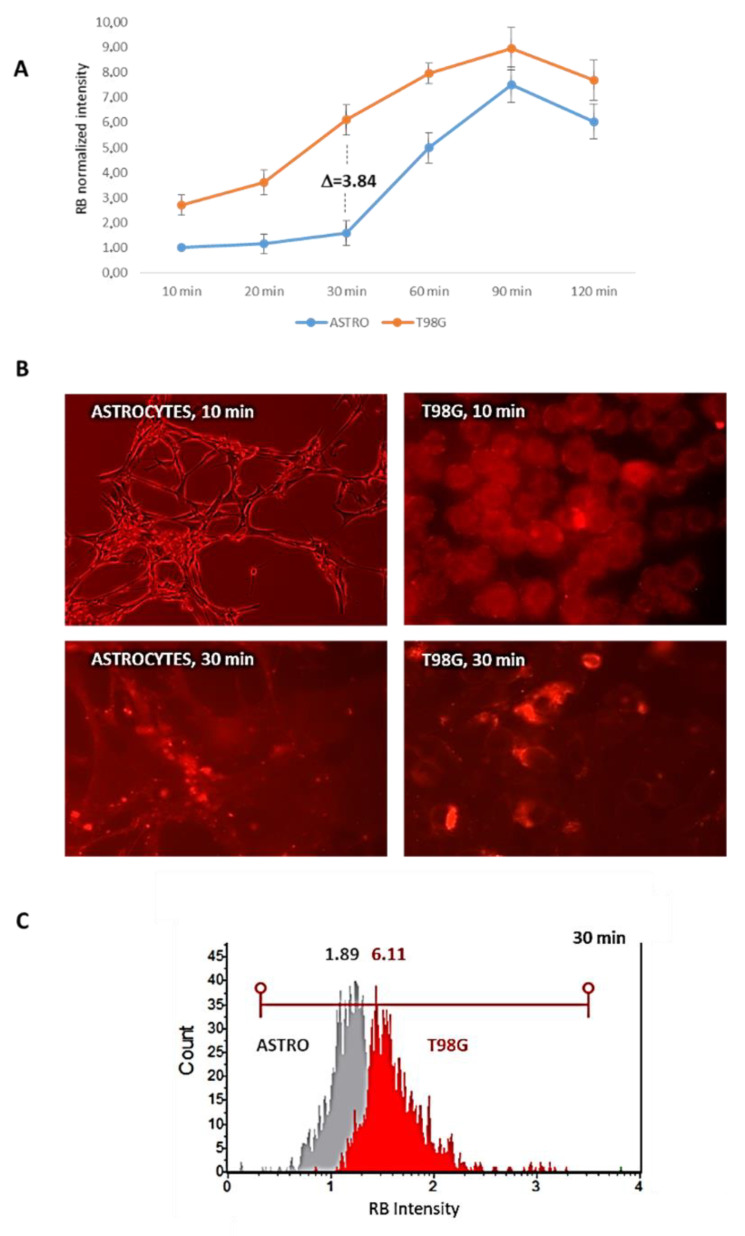
Time kinetics of RB internalization during starvation. (**A**) RB (12.5 μM) was administered during HBSS starvation in normal rat astrocytes and T98G cells and evaluated by Guava Muse Cell Analyzer cytometry (Luminex) at different time intervals (i.e., 10–20–30–60–90–120 min p.t., X axis). Values, obtained in three independent experiments, are normalized to astrocytes at time 0 (reported in Y axis). SD are highlighted as vertical bars. (**B**) RB was visualized in astrocytes and T98G cells at 10 and 30 min p.t. using a Nikon Eclipse TS100 inverted florescence microscope at 20× magnification (RB: G2A blocking filter; exc. = 535/50 nm; dichr. mirr. = 577 nm; barrier = 580 nm). (**C**) Comparison of cytofluorimetric plots of astrocytes (gray) and T98G (red) cells (*n* = 10,000) in RB intensity values at 30 min HBSS starvation.

**Figure 9 cells-11-02502-f009:**
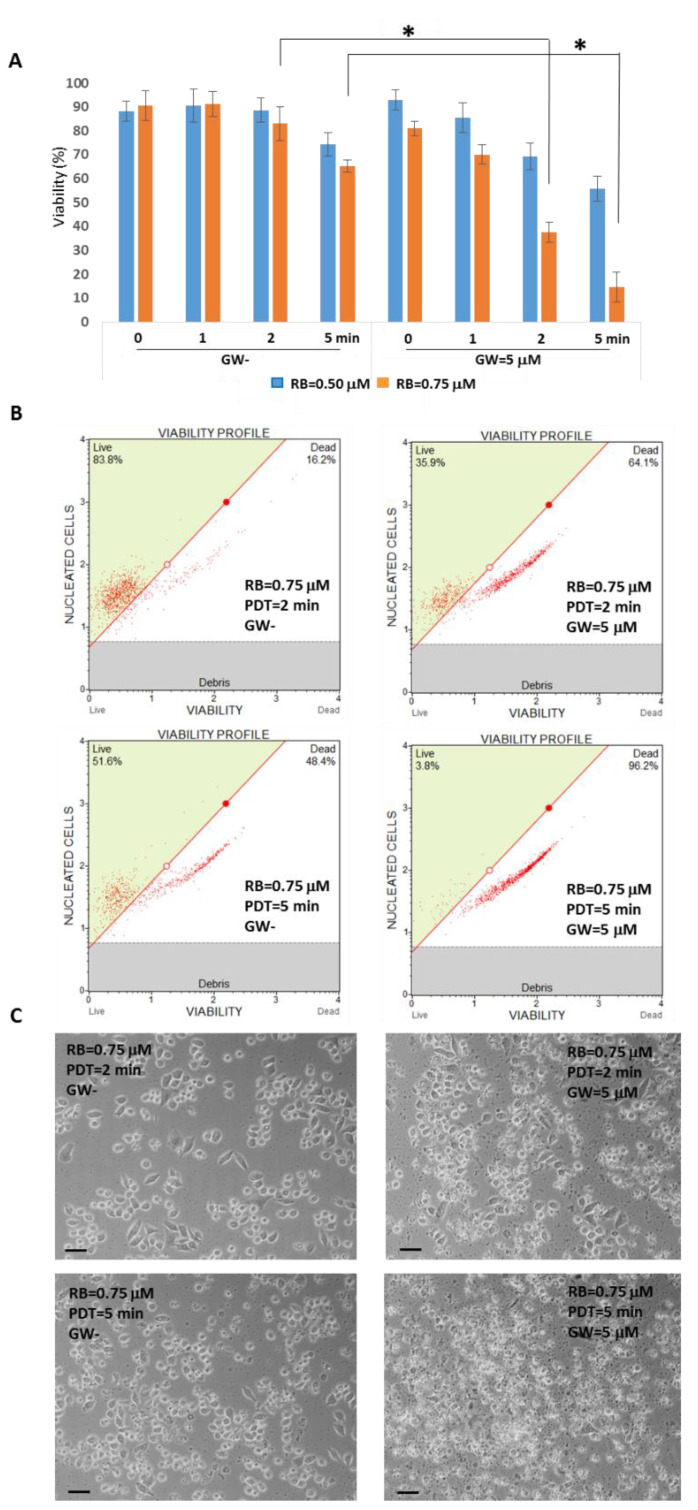
Viability assays following RB starvation, PDT exposition and GW4869 administration in T98G cells. (**A**) Cytofluorimetric viability graph of T98G cells (*n* = 2000) incubated with RB concentrations (i.e., 0.50 and 0.75 μM) for 30 min in HBSS media, with or without GW4869 (GW = 5 μM), combined with different PDT exposition times (i.e., 0–1–2–5 min at 562/576 nm), after 24 h in standard growing conditions (high glucose media). Asterisks indicated statistical significance compared to GW cells (*p* < 0.05, ANOVA one-way; SD are highlighted as vertical bars). (**B**) Examples of cytofluorimetric viability plots and (**C**) optical microscope examinations of RB (0.75 μM), with/without GW4869 and PDT (2–5 min at 562/576 nm). Scale bars = 30 μm.

**Figure 10 cells-11-02502-f010:**
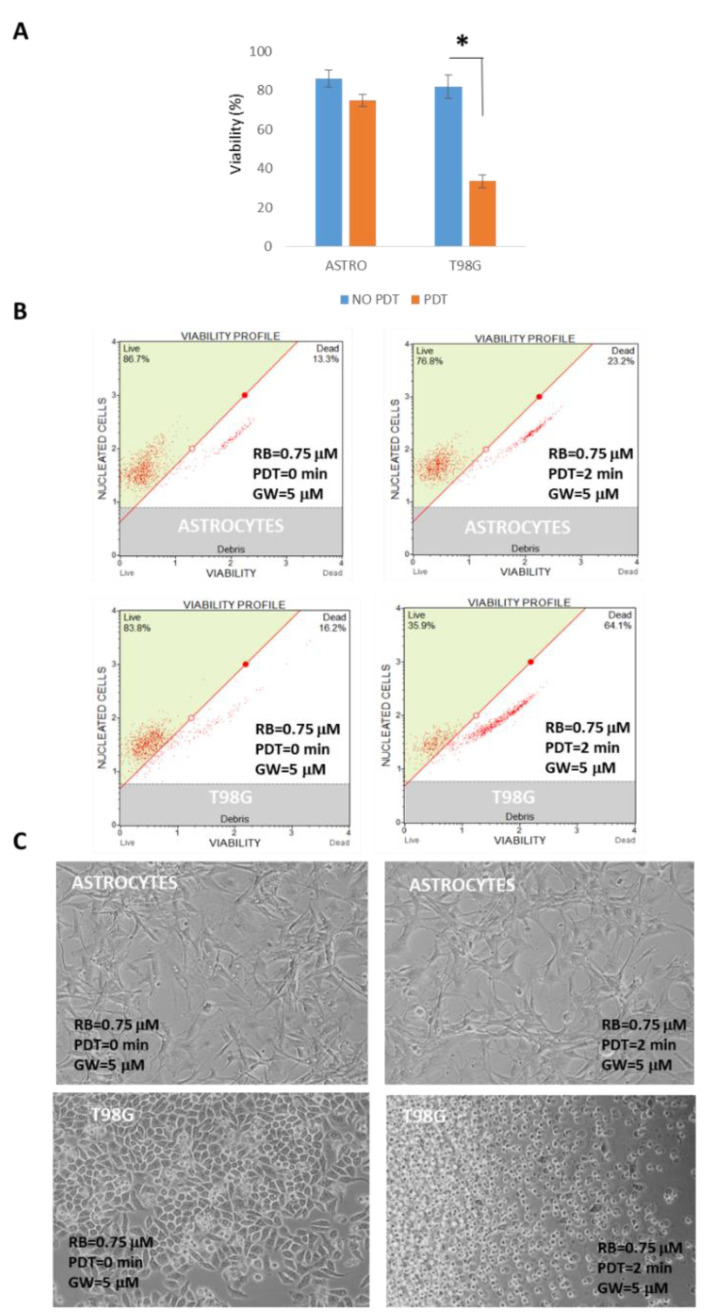
Viability assays comparison following RB starvation, PDT exposition and GW4869 administration in normal rat astrocytes and T98G cells. (**A**) Astrocytes and T98G cells were incubated with RB = 0.75 μM for 30 min in HBSS starvation, which was followed by PDT exposition of 0 or 2 min at 562/576 nm, administration of GW4869 (GW = 5 μM) in reconditioning standard media for 24 h. Asterisk indicated statistical significance compared to not PDT exposed cells (*p* < 0.05, ANOVA one-way; SD are highlighted as vertical bars). Viability graphs and plots were obtained using Guava Muse Cell Count & Viability kit (Luminex) analyzing 2000 cells each of two independent experiments. (**B**) Examples of cytofluorimetric viability plots and (**C**) optical microscope examinations (Nikon Eclipse T100, magnification 10×) of astrocytes and T98G PDT-untreated and -treated cells at 24 h p.t.

**Figure 11 cells-11-02502-f011:**
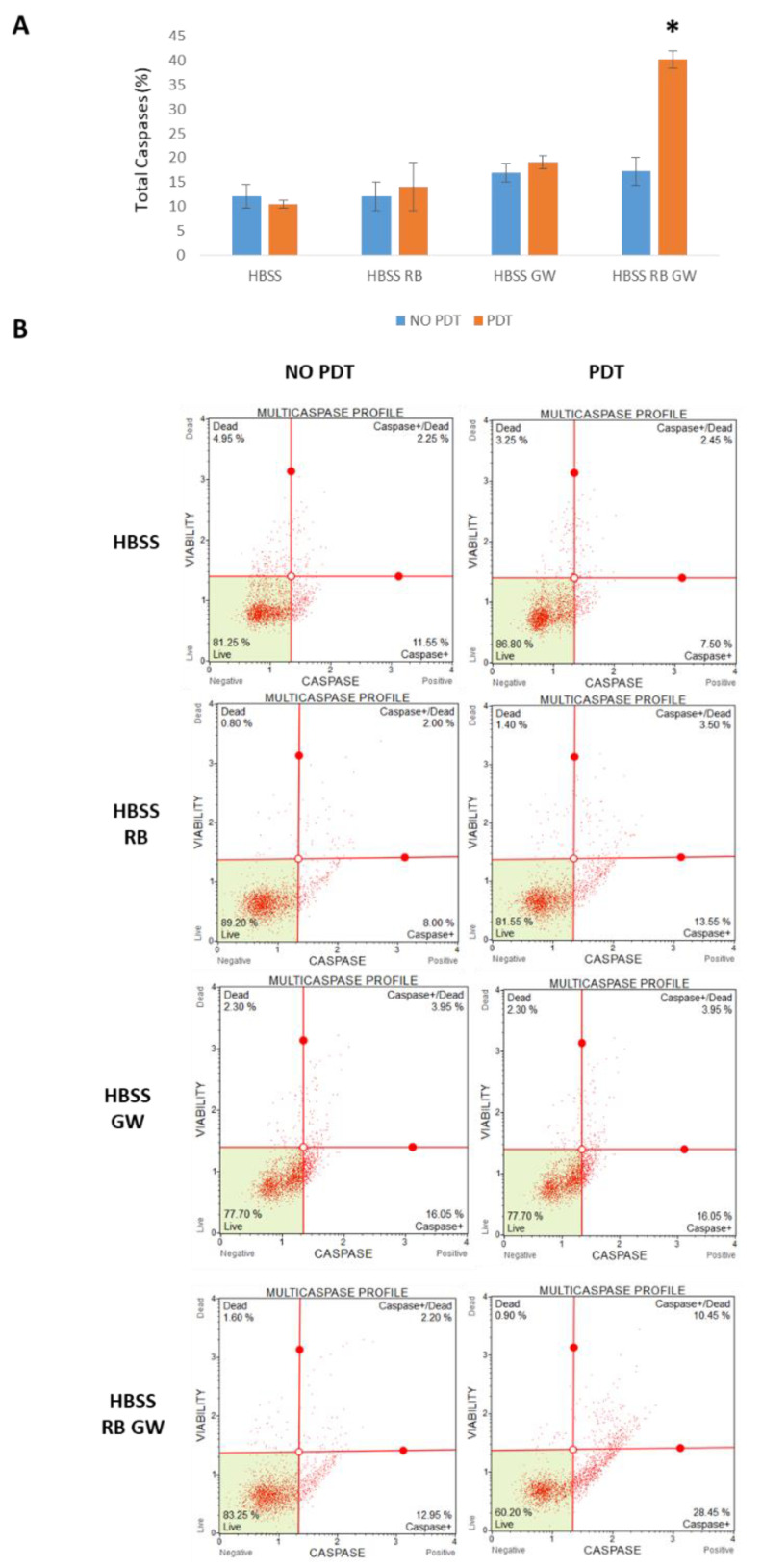
Caspases cytometric analysis of T98G cells following RB starvation, PDT exposition and GW4869 administration scheme. Cells were starved for 30 min in HBSS media, singled treated with RB (0.75 μM) or GW4869 (GW = 5 μM) or with the RB-GW combination, and exposed or not to PDT (2 min at 562/576 nm). Trypsinized cells were finally analyzed after 24 h p.t. in standard growing media for apoptosis induction using the Guava Muse MultiCaspase kit (Luminex). (**A**) The histogram of mean percentages of Total Caspases (i.e., Caspase 1, 3, 4, 5, 6, 7, 8, and 9) positive cells (Caspase + Dead and Caspases + events) of two independent experiments is reported (*n* = 2000 each). Asterisks indicated statistical significance of HBSS, RB, GW and PDT-exposed cells compared to the remaining samples (*p* < 0.05, ANOVA one-way; SD are highlighted as vertical bars). (**B**) Examples of cytofluorimetric MultiCaspase plots.

**Figure 12 cells-11-02502-f012:**
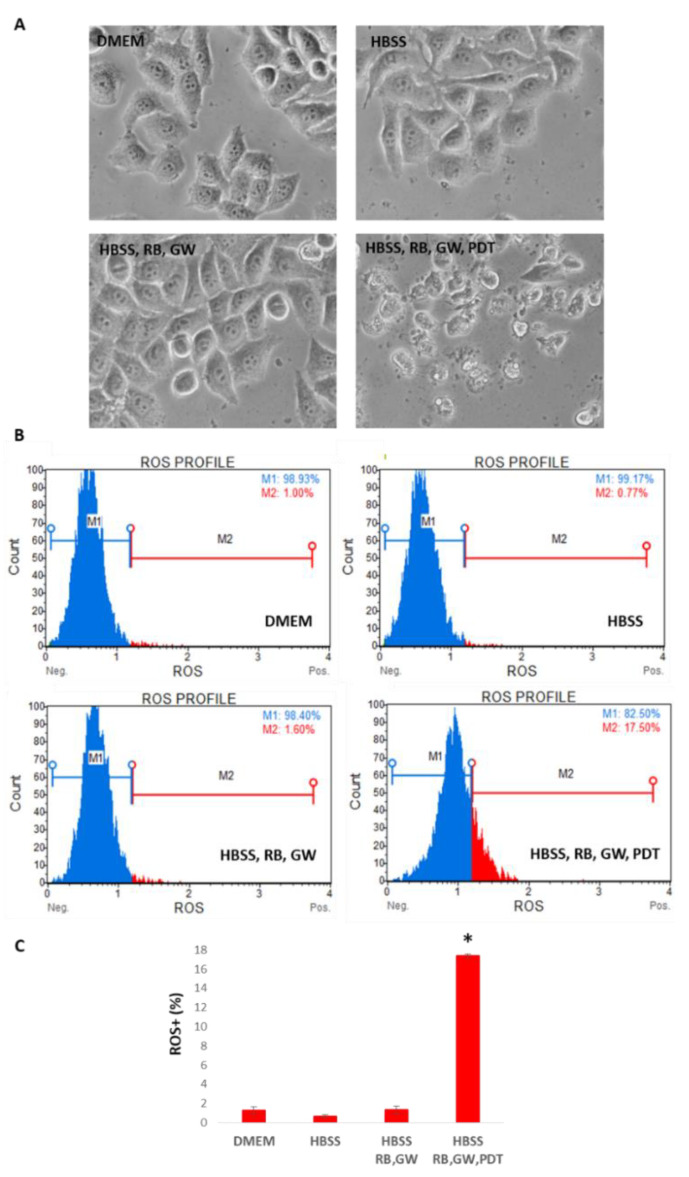
Oxidative stress analysis of T98G cells following RB starvation, PDT exposition and GW4869 administration scheme. Cells were grown in standard DMEM high-glucose medium, starved for 30 min in HBSS media, treated with RB (0.75 μM) and GW4869 (GW = 5 μM), and exposed or not to PDT (2 min at 562/576 nm). (**A**) Cells were visualized using optical microscope at 20× magnification (Nikon Eclipse TS100). (**B**) Trypsinized cells were finally analyzed after 24 h p.t. in standard growing media for ROS^+^ production using the Guava Muse Oxidative Stress kit (Luminex). (**C**) The histogram of percentages of ROS^+^ (i.e., M2 gates) events of two independent experiments is reported (*n* = 3000 each). Asterisk indicated statistical significance comparing HBSS, RB, GW and PDT-exposed cells to the other samples (*p* < 0.05, ANOVA one-way; SD are highlighted as vertical bars). M1 and M2 gates, respectively for ROS^−^ and ROS^+^ events, were established on untreated cells in DMEM high-glucose standard growing condition and adopted for all samples.

**Figure 13 cells-11-02502-f013:**
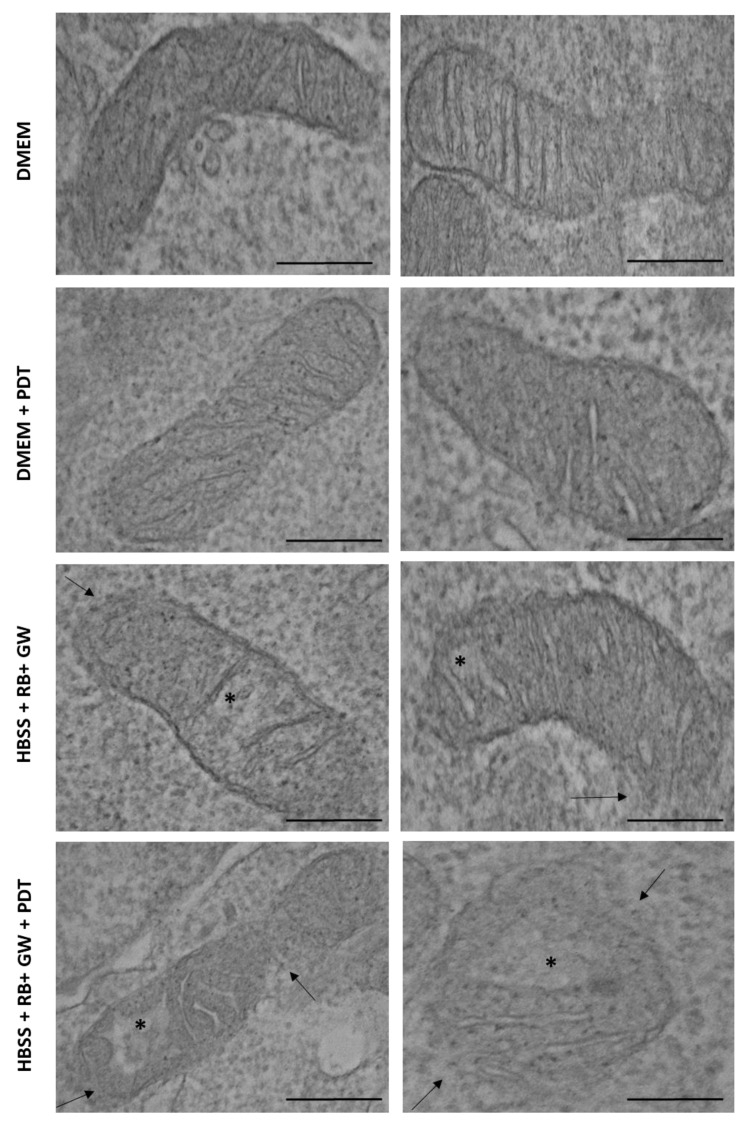
Ultrastructural analysis of mitochondria damage. T98G cells grown in standard DMEM high glucose medium, starved for 30 min in HBSS media, treated with RB (0.75 μM) and GW4869 (GW = 5 μM), and exposed or not to PDT (2 min at 562/576 nm) were then trypsinized and analyzed by TEM after 24 h p.t. Arrows indicated outer mitochondrial membranes damages while asterisks indicated a degradation of the integrity of mitochondria cristae. Scale bars (0.2 μm) are reported.

**Figure 14 cells-11-02502-f014:**
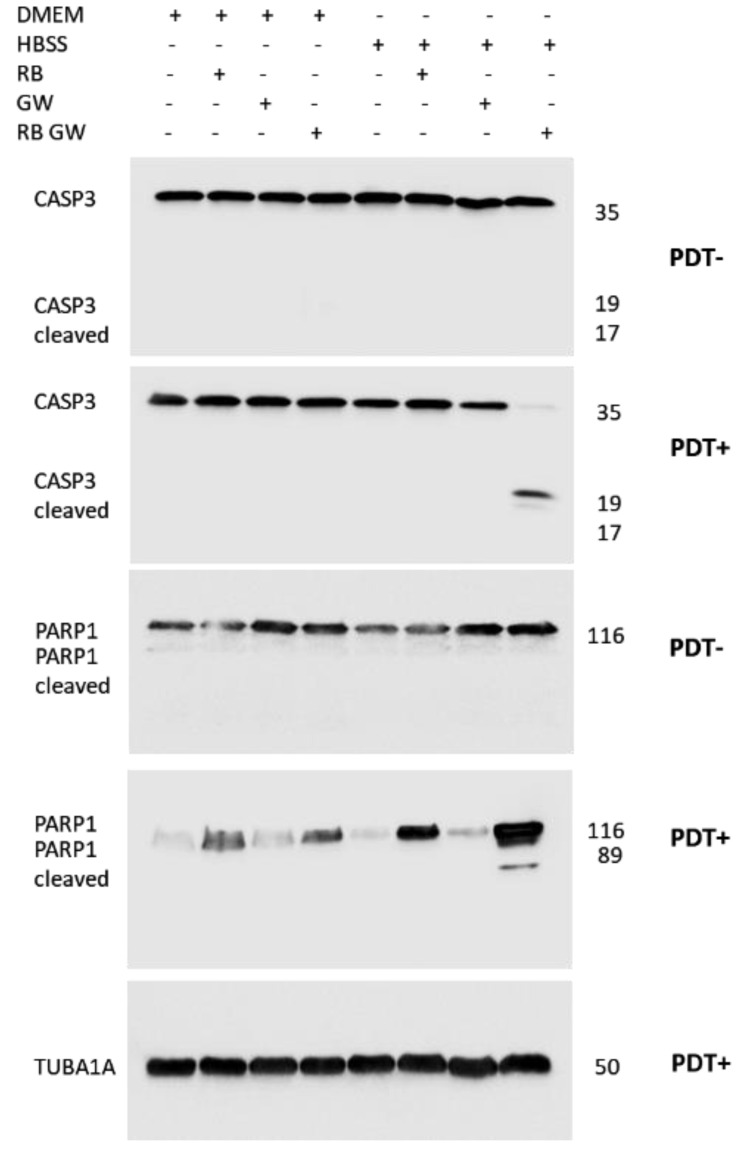
Expression of apoptotic markers in T98G cells following RB starvation, PDT exposition and GW4869 administration scheme. Caspase 3 (CASP3, revealed by anti-Caspase 3 antibody, Cell Signaling, #9662), PARP1 (Cell Signaling, #9532) and **α**-tubulin (TUBA1A, Cell Signaling, #2144) proteins expression in untreated standard growing (DMEM) or HBSS starvation, with/without RB (0.75 μM), GW4869 (GW = 5 μM), PDT stimulation (2 min at 562/576 nm), after additional 24 h p.t. in standard growing conditions. For each sample, 40 μg of total protein extracts was loaded into 12% polyacrylamide gels. All the experiments were performed in duplicates with independent assays. A representative experiment is shown. Molecular weights in KDa are reported.

## Data Availability

Not applicable.

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
