# Peer review of "Enhanced Delivery of Rose Bengal by Amino Acids Starvation and Exosomes Inhibition in Human Astrocytoma Cells to Potentiate Anticancer Photodynamic Therapy Effects"

_cells, 2022, doi:10.3390/cells11162502_

Round 1

Reviewer 1 Report

In this study, the authors establish a simple cell starvation protocol which can lead to a significant improvement in cellular uptake of Rose Bengal (a cell impermeant photosensitizer) by amino acid-starved high-grade human astrocytoma cells. Theoretically, Rose Bengal is a good candidate for photodynamic therapy (PTD) as it displays photosensitization properties with a peak absorption in the visible region of the spectrum (at around 552 nm) and a high quantum yield of production of highly-reactive DNA-damaging singlet oxygen species. The authors further show that the incubation of amino acid-starved astrocytoma cells with Rose Bengal in combination with a small molecule GW4869 exosome inhibitor leads to a maximal intracellular retention of Rose Bengal and improved responses (i.e., cell death) after PTD. My comments for the authors are as follows:

1.    In the experiment illustrated by Figure 7 (i.e., the flow cytometric characterization of exosomes released by starved T98G cells incubated with Rose Bengal), the two fluorochromes used in the ImageStream analyses (i.e., Rose Bengal and PHK26) have very similar spectral properties. The Ex/Em spectra of Rose Bengal in aqueous solutions are 552 nm/567 nm, while the Ex/Em spectra of PKH26 dye are 551 nm/567 nm. The use of this combination of fluorochromes with overlapping spectra makes the interpretation of the presented results very difficult. 

2.    Similarly, the ROS detection dye (i.e., dihydroethidium) employed by the authors in their FACS analysis experiment illustrated by Figure 12 has an overlapping spectra with Rose Bengal and makes the interpretation of that dataset equally difficult. 

3.    The authors show that the combination of starvation, incubation with high nanomolar concentrations of Rose Bengal followed by 5 μM of GW4869 and PTD leads to increased levels of cell death in human astrocytoma cells. However, in a study conducted with breast cancer cells by Luberto et al. (DOI: 10.1074/jbc.M206747200 ) it was shown that the small molecule neutral sphingomyelinase inhibitor GW4869 can, in a dose-dependent manner, “significantly protect from cell death as measured by nuclear condensation, caspase activation, PARP degradation, and trypan blue uptake”. These seemingly cell protective effects by GW4869 could further complicate the conclusions of the present study. I would like the authors to comment on this. 

4.             Lastly, it is very unclear how translatable the findings of the present study are. It is highly unlikely that the starvation conditions used by the authors in their cell culture experiments could be readily translated in a glioblastoma patient in order to improve the tumor bioavailability of Rose Bengal in a similar manner. The authors present no in vivo experimental data (especially biodistribution studies) with Rose Bengal and GW4869 (+/­ PTD) in xenograft models of human astrocytoma in rodents. Therefore, the authors should address these translational limitations of their study in the Discussion section  of the manuscript and further address the envisioned practical applications of their findings.  

Reviewer 2 Report

Title

Enhanced Delivery of Rose Bengal by Amino Acids Starvation and Exosomes Inhibition in Human Astrocytoma Cells to Potentiate Anticancer Photodynamic Therapy Effects.

Concise Summary

The authors aim to investigate photodynamic therapy (PDT) is a promising anticancer strategy in glioblastoma (GBM) based on cytotoxicity induced by the activation of photosensitizers as Rose Bengal (RB) by protoinducers. In order to increase RB uptake in human astrocytoma cells and to interfere with RB extracellular release exocytosis by extracellular vesicles, the authors have developed an experimental pharmacological treatment based on the combination of short-term amino acids starvation and GW4869 exosomes inhibitor. Thus, the authors postulate that this pharmacological approach induced cell culture astrocytoma viability through the activation of apoptotic pathways. It is concluded that this approach improves the effectiveness of PDT-based treatments and could be important strategy for chemotherapeutic drugs in tumor cells.

Major Comments

It has been shown in this article that the incubation of GBM cells with RB resulted in development of tumor cell apoptosis due to photoactive RB. It is possible that RB could have a practical application in PDT through toxicity produced by introduction of acetate groups increases in tumor cells. The results presented in this article are very preliminary ones and obviously need validation by in vivo experiments. Other consideration is the adverse effects in case of these results could be extrapolated to a clinical setting. Moreover, other procedures to improve the delivery RB through nanoparticle technology provides other interesting opportunities to improve the bioavailability of photosensitizing molecules while reducing side-effects.

Methodologically, the study involves a combination of cell culture and diverse procedures to evaluate RB uptake, analysis of fluorescent staining of EVs, RB cellular internalization, analysis of proteins expression, cell viability and morphology of normal rat astrocytes and astrocytoma cells (two cell lines: U373-MG and T98G), The methodology has been applied correctly.

Questions:

1.The authors state (lines 249-50) that examinations and flow cytometry internalization measurements were performed on the lowest and highest RB doses. Could the authors explain which is the rationale of the dose selection?

2. It is said that (lines 287-90) “Among the investigated cell lines …, the most significant normalized difference between RB administration in standard condition and in amino acids starvation was reported for T98G.” It has been told in the article that T98G cells are the most effective in RB induced uptake, after amino acids deprivation and exosomes inhibition. It would be interesting to know which are the reasons for the different behavior obtained in the tested GBM cell lines. Could the authors explain the reason of these differences?

3. The authors used rat normal astrocytes in the experiments, but not human ones. Which is the reason for this decision?

4. The proposed PDT-based treatment revealed morphologic mitochondrial alterations. These are very interesting and complete results, but a reference about ultrastructural mitochondrial observations in normal human astrocytes could be needed.

5. In the article it is concluded that this approach improves the effectiveness of PDT-based treatments and could be important strategy for chemotherapeutic drugs in tumor cells. However. from a practical point of view, the proposed combination of amino acids deprivation and EVs inhibition pose obvious practical difficulties to be implemented. The results presented in this article are based only in an in vitro experiment and obviously need validation. Could the authors explain how this treatment procedure could be applied in an in vivo experimental model?

Minor Comments

1. References should not be included in Results section (lines 338, 355).

2. The double numbering in References section should be corrected.

Conclusion

The article addresses a relevant clinical issue through an interesting experimental model. I consider that the article is original and valuable. It is generally well written and the considerations about the issue are consistent, but it is needed more information to be sure about the relevance of the possible future clinical impact.  

Round 2

Reviewer 1 Report

The authors provided a satisfactory answer (and solution) that certainly aids with the interpretation of their data obtained from ImageStream and FACS analyses which employed fluorochromes with overlapping spectra. I would also like to thank the authors for their detailed answer regarding the in vivo applicability of their in vitro approach. I am very familiar with the studies conducted by Dr. Longo, but I am not entirely convinced that an effective tumor amino acid starvation could be achieved  in vivo (specifically in a cancer patient) strictly via dietary means such as the administration of protein-free meals. The strict dependence of cancer cells on certain amino acids (such as glutamine), as a critical source of nitrogen needed for biosynthetic pathways (especially for nucleotide synthesis), dictates that these amino acids will be nonetheless obtained from other sources when not provided in the diet, such as from the breakdown of patient's own muscle tissue, etc. For this reason, the ability to manipulate the amino acid pools in tumors via dietary means is probably more complicated than it seems, unless additional pharmacological means are also employed.  Ultimately, the in vivo validation of authors' hypothesis that the restriction of dietary proteins will help the tumor uptake of RB in tumor bearing animals needs to be tested empirically. However, the inclusion of such in vivo biodistribution data would have significantly benefited the present study and  increased the confidence in the feasibility of the approach proposed by authors.